# Contribution of Microhomology to Genome Instability: Connection between DNA Repair and Replication Stress

**DOI:** 10.3390/ijms232112937

**Published:** 2022-10-26

**Authors:** Yuning Jiang

**Affiliations:** Department of Radiation Oncology, University of Virginia, Charlottesville, VA 22903, USA; xdk4xz@virginia.edu; Tel.: +1-434-996-1963

**Keywords:** microhomology-mediated end joining (MMEJ), DNA repair, DNA end resection, replication stress, synthetic lethality, cancer therapy

## Abstract

Microhomology-mediated end joining (MMEJ) is a highly mutagenic pathway to repair double-strand breaks (DSBs). MMEJ was thought to be a backup pathway of homologous recombination (HR) and canonical nonhomologous end joining (C-NHEJ). However, it attracts more attention in cancer research due to its special function of microhomology in many different aspects of cancer. In particular, it is initiated with DNA end resection and upregulated in homologous recombination-deficient cancers. In this review, I summarize the following: (1) the recent findings and contributions of MMEJ to genome instability, including phenotypes relevant to MMEJ; (2) the interaction between MMEJ and other DNA repair pathways; (3) the proposed mechanistic model of MMEJ in DNA DSB repair and a new connection with microhomology-mediated break-induced replication (MMBIR); and (4) the potential clinical application by targeting MMEJ based on synthetic lethality for cancer therapy.

## 1. Genome Instability and Cell Evolution

Cell growth and proliferation need to be tightly coordinated to ensure the preservation of genome integrity and to promote faithful genome propagation. Efficient and accurate DNA replication and repair is key for the faithful duplication of chromosomes before their segregation. Interestingly, a recent study reported a new form of cell division in zebrafish without the involvement of DNA replication [1]. A faithful coordination of DNA replication with DNA repair and cell cycle progression leads to genome integrity during cell divisions and avoids chromosome rearrangements. Two types of elements contribute to instability: suppressors, including replication, repair, and S-phase checkpoint factors; and chromosomal sites, which act as regions of noncoding DNA as hotspots of instability and include fragile telomere sites and highly transcribed DNA sequences. These factors generate genomic instability that leads to pathological disorders and is also crucial for evolution [2]. During the cell cycle, DNA is most vulnerable to process replication during the S phase, and the replisome must overcome obstacles. When DNA is not properly processed, genome integrity must still be maintained. Thus, eukaryotic cells have developed checkpoint functions that are constantly monitoring the integrity of the DNA and serve to coordinate replication with repair, chromosome segregation, and cell cycle progression.

DNA double-strand breaks (DSBs) are the most toxic form of DNA damage. DSBs can occur due to both endogenously arising compounds, such as reactive oxygen species, and exogenous exposure to environmental factors, such as mutagenic chemicals and radiation [2]. To minimize the effect of this damage, cells have evolved various DNA repair mechanisms. Unfaithful repair of DNA DSBs leads to genome instability (GIN), in which cells may survive but generate genetic mutation, deletion, duplication, and/or the accumulation of chromosomal rearrangements [3]. The consequences of these events can lead to cancer, disease progression, and therapy resistance. Accurately repairing DSBs is important for cell division, and the progress is controlled during the whole cell cycle. Replication-associated DNA breaks can be generated in several ways, such as encountering a single-strand DNA nick, which results in the discontinued synthesis of nascent strands and leads to DSBs [4]. This event can occur due to replication stress-induced compounds, which is a different process from that of radiation-induced blunt ends of DSBs. If the nick is on the leading strand, the DSB would be one-ended and could promote the restart of synthesis by break-induced replication (BIR). BIR usually occurs after inducing a single-strand nick. Replication fork progression or leading-strand synthesis can be blocked, and leading-strand and lagging-strand synthesis are uncoupled [5,6]. A lesion blocks lagging-strand synthesis and creates an ssDNA gap or a DSB if the lesion is a nick. Lesions that block leading-strand synthesis can be bypassed by the replication fork, and replication can restart downstream and leave ssDNA gaps behind [5]. ssDNA gaps can be repaired by homologous recombination or error-prone translesion synthesis (TLS) [7]. Homologous recombination (HR) is one of the major pathways to repairing DSBs and mainly functions in the S phase, but it also functions in G2 and mitosis. It is initiated by DSBs but also addresses replication fork stalling-induced DSBs and acts as an alternative process for ssDNA gaps repair [8]. Compared with another major DSB repair pathway, NHEJ, which was thought as potentially error-prone for introducing insertions or deletions during the DSB end processing, HR was thought to be a “safer” repair option, which uses the exact same genomic information as a template for repairing the DSB [9]. A recent study suggested that the regulation of repair at stalled replication forks differs from that of a conventional DSB. Another pathway, single strand annealing (SSA), mediates error-free repair at stalled forks by suppressing tandem duplications at sites of aberrant replication fork restarts [10]. Different from the above pathways, canonical non-homology end joining (C-NHEJ) occurs throughout the cell cycle and repairs DSBs faster, but it is more active in the G1 phase [11]. There is still less evidence about microhomology-mediated end joining (MMEJ), which is also active in the S phase, and its role in replication fork stalling. Microhomology-mediated template switching processing of stalled replication forks is possible [10]. Recent studies have raised the central question of whether the replicative helicase CMG (CDC45-MCM2-7-GINS) participates in replication restart after fork collapse. A new study observed the fate of the replication fork after collision with strand-specific nicks. The CMG helicase is lost from the DNA but performs a different function with single-end DSBs. Short resection and gap fills occur on the leading strand [12]. Fen1, which is functional on MMEJ, participates in nascent DNA synthesis [13,14]. It has been revealed that the GINS complex (consisting of four proteins: Sld5–Psf1–Psf2–Psf3, and named for the Japanese “go-ichi-ni-san” [15,16]) molecules move at the leading edge of growing Fen1 tracts by imaging fluorescence Fen1 signal as nascent DNA synthesis [17], to demonstrate that GINS molecules travel with an active replication fork [12]. It is likely possible for ssDNA nicks to undergo short resection of the gap filled in by POLQ, which would mediate microhomology to repair bypass replication fork stalling. When DSBs occur in interphase, cells are able to arrest the cell cycle and repair breaks before entering mitosis. However, when DSBs occur during mitosis, the cycle can no longer be arrested, and cell division is completed instead of repairing the DNA damage [18]. These observations suggest that important factors involved in not only HR but also MMEJ participate in replication fork regulation. Future research could detect how the effect of MMEJ on the CMG complex triggers replication stress. In brief, the mechanisms of replication fork stalling and DSB repair are tightly related, and their interaction is essential for genome integrity and cell evolution (Figure 1).

## 2. MMEJ and Genome Instability

There are two major pathways to repair DSBs, one of which is homologous recombination, which is a largely accurate pathway that recruits RAD51 and BRCA2 and uses sister chromatids as templates for repair following DNA replication. There are several types of HR, including gene conversion (GC), synthesis-dependent strand annealing (SDSA), and break-induced replication (BIR). Another DNA DSB repair pathway is single-strand annealing. They all initiate repair with the resection of DSB ends by a 5′-to-3′ exonuclease to produce long 3′-ended, single-stranded DNA (ssDNA) tails [19]. C-NHEJ requires the KU70/80 heterodimer to bind to DSB ends, the DNA-dependent protein kinase catalytic subunit (DNA-PKcs), X-ray repair cross-complementing 4 (XRCC4), DNA Ligase 4 (LIG4), and XRCC4-like factor (XLF) to efficiently ligate the breaks, with occasional deletion or insertion of DNA information and using 1–4 nucleotide homology or non-complementary base pairing [20,21]. A recent study showed that C-NHEJ is critical for end joining using 0–2 nt terminal microhomology but is relatively dispensable for end joining events involving 3–4 nt [22]. In the absence of Ku70/80 or DNA Ligase IV [23], robust alternative NHEJ activity is observed in yeast and especially in mammals [23,24]. A new system to detect DSB repair by following the expression of Cas9 and sgRNA targeting intrachromosomal fluorescent reporter discriminates between high-fidelity (HF) and error-prone NHEJ, and it has been observed that HF-NHEJ was strictly dependent on DNA Ligase IV, XRCC4, and XLF, components of C-NHEJ [25].

MMEJ is a highly mutagenic DSB repair pathway that leads to deletions at junctions and is associated with chromosome translocations and rearrangements, as well as telomere fusion events [26,27]. Sequence analysis of telomere fusions in human malignancies identified microhomologies (MHs) and deletions that extended into the adjacent nontelomere DNA [28,29]. During the process, telomeres suppress MMEJ by shelterin, which binds to telomeric TTAGGG repeats as a protective protein complex [30,31]. Deleting the six-subunit shelterin complex that also lacks Ku, which is the major factor in promoting NHEJ, would lead to MMEJ activity [30]. Studies have also indicated that even when HR and C-NHEJ are active, MMEJ is still used with appreciable frequency to repair DSBs, especially in mammalian cells [32].

Important proteins that promote MMEJ include poly ADP ribose polymerase (PARP1), DNA polymerase theta (encoded by the *POLQ* gene), and ligase III. POLQ is an essential protein for promoting MMEJ and is reported to suppress recombination [33]; thus, it is thought that MMEJ functions through Pol theta-mediated end joining. However, POLQ is important for repair events using 4–6 nt (but not more than 18 nt) flanking repeats, which are at the edge of the break, as well as oligonucleotide microhomology-templated (12–20 nt) repair events requiring nascent DNA synthesis [34]. Knocked down POLQ and ligase III dramatically decrease chromosome fusion (should be telomere fusion) [33]. POLQ is a DNA polymerase containing an N-terminal helicase domain, which includes both ATPase and DNA unwinding activities. ATPase activity is required for both the suppression of HR by binding RAD51 and inhibiting its assembly along RPA-coated ssDNA and stimulating MMEJ [35,36]. The long, unstructured central region of POLQ contains a RAD51 interaction motif. Protein interaction studies have shown that amino acids 847–894 were both necessary and sufficient for RAD51 binding. The interaction of the central domain of POLQ with RAD51 inhibited RAD51-ssDNA nucleoprotein filaments. Thus, POLQ displaces RAD51, inhibits HR, and promotes MMEJ [36,37]. This could demonstrate the competitive nature between MMEJ and HR occurring at the level of long-range ssDNA binding, which requires multimers of POLQ that include the central region of the protein [24]. However, further studies will be needed to investigate whether structural changes or ATP activity changes in POLQ affect the RAD51 filament stabilization effect. The C-terminal domain of POLQ contains a DNA polymerase domain that performs gap filling, HR inhibition, and microhomology annealing fill-in synthesis at DSBs [36]. The purified POLQ polymerase domain alone is active on short ssDNA and short 3′ overhangs. Longer ssDNA substrates require both the POLQ N-terminal helicase domain and the C-terminal polymerase domain (Figure 2). POLQ can prime DNA synthesis from nonoptimal base pairing, leading to the introduction of insertions at the break sites. But with the 3′ flaps acting as a template, it is highly error prone [38,39,40]. It is unknown whether microhomology can be promoted with a shorter microhomology base pair without POLQ. Unlike SSA, MMEJ is a Rad52-independent pathway in yeast. Some research in yeast has shown that MMEJ required 8–20 nt microhomology [41]. Rad52 is dispensable for microhomologous sequences <14 nt in yeast, which could separate MMEJ and SSA according to Rad52 recruitment [42]. Microhomology-mediated end joining (MMEJ) can be one type of alternative end joining, which relies on 5–25 nucleotide microhomologous sequences on either side of the DSB to repair the breaks in mammalian cells [43]. Interestingly, for end-joining events involving 3–4 nts of terminal microhomology, MMEJ and C-NHEJ can still occur without the most relevant genes involved [22]. Microhomology of more than 10 nucleotides is needed to recruit RAD52. Therefore, I propose that most likely 5–10 microhomology-mediated end joining is POLQ-mediated MMEJ but does not overlap with SSA. Furthermore, whether POLQ functions in other repair pathways, such as cross-link repair, is still unknown, and it will be interesting to uncover the competition or interaction between crosslink repair, MMEJ, and the replication repair mechanism through further studies.

## 3. Initial DNA End Resection Promotes MMEJ

Unlike C-NHEJ, which requires very little or no end resection but often processes small deletions at noncompatible DSB ends, MMEJ relies on short-range end resection to initiate the repair process. Initial short-range end resection, even of fewer than 20 bp, at the DSB end, is enough to promote MMEJ, in contrast to HR and SSA, which both need long-range extensive end resection [32,44]. In yeast, DNA end resection is started by initial end resection (100–200 bp) followed by extended end resection, which is promoted by BLM, DNA2, and EXO1, which favor HR and SSA [45]. Both the MRE11–RAD50–NBS1 complex (MRN) and CtIP are required to promote the initial resection of MMEJ in mammalian cells [46,47]. In particular, MRE11 nuclease activity is required for MMEJ involving short-range end resection. Initial resection occurs in the S phase since cyclin-dependent kinases (CDKs) phosphorylate several key factors that activate resection and weaken the impediments to end resection [48] (Figure 3). In noncycling cells, DSBs may favor C-NHEJ, as DNA end resection will be reduced dramatically [49]. In cycling cells, another important factor in promoting end resection is CtIP, and the role of CtIP in initial resection is still not fully understood. Some studies have shown that phosphorylation of CtIP Thr847 by CDKs is certainly involved in promoting DNA end resection [50]. CDKs initiate DNA end resection by stimulating MRE11 endonuclease activity [51]. Notably, as the co-factor of MRE11 in initiating end resection, mutating Thr847 reduced the occurrence of both MMEJ and HR, which suggested that the function of CtIP with phosphorylated Thr847 is important for the initial short-range of end resection to activate MMEJ [32]. CtIP interaction with NBS1 and CDK-dependent phosphorylation facilitate ATM to phosphorylate CtIP [52]. Both CDK and ATM phosphorylation of CtIP are important for end resection [50,53,54]. 53BP1 localizes to DSBs in G1 phase cells and promotes NHEJ and is required for telomere fusion and results from deprotection of chromosome ends [55]. BRCA1 counteracts 53BP1 in the S/G2 phase to promote end resection of HR. Studies have found that loss of 53BP1 restores end resection in BRCA1 mutant cells [56]. However, the BRCA1 interaction with CtIP is not essential for resection but affects resection speeds [57]. A recent study reported that resection-dependent C-NHEJ also occurs in the G1 phase as an inducible process during which PLK3 phosphorylates CtIP, mediating its interaction with BRCA1 and promoting resection. In the G1 phase cells, DSBs including those localizing to heterochromatic regions or additional lesions at the DSB site, undergo resection prior to repair by C-NHEJ through this mechanism but not alt-NHEJ [58]. MMEJ as a type of alt-NHEJ (see below) could also happen in the G1 phase. Previous studies have shown that in the G1-phase, 53BP1 supports sequence deletion during MMEJ consistent with the role of 53BP1 in end resection, but it is only observed in the presence of functional BRCA1 [59]. PLK1 targets CtIP phosphorylation at serine 327 to promote MMEJ and inactivate the G2/M checkpoint [60]. A well-studied protein, retinoblastoma protein (RB), was recently identified as a new factor in MMEJ and potentially regulates CtIP function-mediated DNA end resection [61]. However, there is still much to learn about the detailed mechanism of regulating resection initiation and resection speed, which could both be important for MMEJ regulation. The structural function of CtIP, such as the role of dimerization and the interaction between CtIP and other cofactors, in the regulation of MMEJ, is crucial information.

## 4. Synthetic Lethal Role of MMEJ for Other DNA Repair Pathways

MMEJ was first discovered as a type of alt-NHEJ when the NHEJ protein Ku family was defective in *Saccharomyces cerevisiae* [62]. It has been detected as a Pol theta-mediated microhomologous end joining in most forms of life, including bacteria, yeast, flies, worms, plants, zebrafish, and mammals [63,64,65,66,67]. Early evidence of MMEJ in mammalian cells was obtained from the analysis of class-switch recombination (CSR) and V(D)J recombination in NHEJ-defective B cells [27]. Defective Ku, Xrcc4, Lig4, DNA-PKcs, and Artemis all lead to MMEJ engagement and an elevated length of microhomology [68]. It could also be possible to block the ATM phosphorylation of DNApk or disrupt DNApk activity, but without full-length DNApk depletion, elevated MMEJ is still observed. A newly unraveled alt-end joining mechanism was not initially detected in either *LIG4^−/−^* or *XRCC4*^−/−^ mouse B cells, and microhomologies were recovered at all CSR junctions, implying that the mechanism could be repressed by the Ku family. However, no reduction in CSR was found in *POLQ*^−/−^ mice, either in NHEJ-proficient or NHEJ-deficient mice [39]. In addition, a later study also found that the mutated form of RAG endonuclease activated MMEJ and decreased V(D)J recombination in both wild-type and DNA-PK-deficient cells [69]. Two types of alternative end-joining pathways of DSB repair have been identified in NHEJ-defective cells. One is POLQ-mediated end joining. MMEJ relies on preexisting microhomologies around the break and relies on ligase III for annealing. Another alternative pathway does not require preexisting microhomologies and may instead rely on ligase I. One proposal is that microhomologies are nevertheless generated by a polymerase activity operating on one DNA end [26]. Ligase I may only function in the absence of ligase III as a backup ligase in mouse cells. Both ligase I and ligase III are repressed by NHEJ [70]. It was thought that MMEJ is a backup pathway for NHEJ; however, when NHEJ is active, there is still evidence showing MMEJ function, which suggests the coexistence of MMEJ and NHEJ [68]. Blunt DSB ends could be generated from filled-in or degraded ends with short 5′ overhangs. DSB ends can be processed by polymerases that add nucleotides and generate microhomology. Short microhomology could stabilize the blunt ends and cause short insertion–deletion (indel) mutations. When DSB ends are processed into 3′ ssDNA to be substrates for POLQ, the position of microhomology that flanks the DSB would govern the size of the deletion mutation [71].

In addition to NHEJ, the synthetic lethal relationship between HR and MMEJ was also an essential discovery, which provided convincing evidence that homologous recombination-deficient cancer depends on POLQ-mediated MMEJ [37]. This finding challenged the traditional balance between HR and NHEJ and explained the new DSB repair pathway choice between relatively accurate repair and more mutagenic repair. However, the mechanism of that pathway shift is still unknown. Briefly, the potential competition mechanism between MMEJ and other DSB repair pathways is shown as (Figure 4). The insufficiently extensive DNA resection for HR but initiation of resection sufficient for MMEJ has been observed. The RPA complex, as an important step for extensive DNA resection, could suppress MMEJ [72]. Interestingly, RPA phosphorylation inhibits DNA resection through inhibition of BLM, which is the helicase together with EXO1 and DNA2 nucleases, to promote extensive resection of HR. RPA32 phosphorylation induces physical interactions with RPA70N, which interacts with BLM to form a loop of resection during the DNA damage response [73]. These RPA functions should balance the extensive resection in HR but suppress MMEJ. In addition, the helicase domain of POLQ counteracts RPA to promote MMEJ [74]. Recently, it has been reported that the inhibition of POLQ function also induces toxic RAD51, which replaces RPA to continue the single-strand invasion of the template of HR, and could also block terminal resection in HR and cause a shift to MMEJ [75]. The detailed mechanism by which POLQ induces toxic RAD51 is still unknown. It might be possible for POLQ function to induce unstable RAD51 filament or degrade RAD51 filament binding to ssDNA. This may also be due to the speed of resection initiation supporting MMEJ even faster than cell DSB repair by HR. The important gene functions regulating either resection initiation or the speed of resection initiation for these two pathways are still unknown.

Break-induced replication (BIR) is another DSB repair pathway that leads to genomic instability to repair one-ended DSBs. Yeast model studies described BIR synthesis that was carried out by migrating bubble and shows the conservative inheritance of newly synthesized DNA [76]. BIR leads to genome instabilities probably by driving more chromosome translocations by switching the extending DNA strand from its template sequence to another homologous template during DNA replication. Chromosome translocation during replication can be observed in human cancers [77]. BIR is initiated when only one broken end is available for strand invasion. The invasion drives the DNA repair synthesis process via a migrating bubble with leading and lagging strand synthesis and leads to the conservative inheritance of newly-synthesized DNA. BIR is a unique HR mechanism employed in the situation where a single end of DSBs acts independently. It may occur when one side of the break fails to engage with a homologous sequence or when the two ends find different homologous templates [76]. BIR arises to promote repair when DSB manifests as a “one-ended break”, which can occur due to replication through a DNA lesion that results in fork stalling and collapse, or telomere erosion that exposes a single DNA end [78]. BIR is also processed by 5′-3′ resection of one DSB end, invades the homologous template, and initiates synthesis that can copy >100 kb of the template until the end of the chromosome. The breaks of BIR are formed by collapsed replication forks or eroded telomeres that lack telomerase in pathways known as alternative lengthening of telomeres [79]. Instead of utilizing a canonical replication fork, BIR is driven by a migrating D-loop bubble, which leads to the conservative inheritance of newly synthesized DNA and is associated with a high frequency of mutagenesis [80,81]. In addition, during BIR, the leading and lagging strands are synthesized in an asynchronous manner, leading to the accumulation of long ssDNA regions stabilized by RPA and RAD51 nucleofilaments [82]. Recent data have shown that BIR can be induced by replication stalling at chromosome common fragile sites (CFSs) [83]. The nuclease activity of MUS18 promotes POLD3-dependent DNA synthesis at CFSs during early mitosis, which indicates that the BIR-like process is dependent on the POLD3 and POLD4 subunits of Pol delta in mammalian cells [84]. In yeast, Pol32, a nonessential subunit of the DNA polymerase delta complex, increases its processivity and, to a lesser degree, its dependence on the Pif1 helicase [85]. BIR synthesizes significantly more DNA than what is synthesized during the repair of a two-ended DSB, and newly synthesized ssDNA is released behind the D-loop. During this process, the other end of the break has no opportunity to reanneal. During the S phase, mismatch repair (MMR) corrects DNA synthesis errors introduced during the BIR process [86]. Mutations that accumulate during the synthesis of the invading strand are made permanent by the synthesis of the second strand [87]. Interestingly, it has been reported recently that analogous to CFSs, fragile telomeres in BLM-deficient cells involved DSB formation, and the BIR of telomeric DSBs competed with PARP1-LIG3 and XPF-dependent alt-NHEJ, which did not generate fragile telomeres [88]. This finding indicates that the new contribution of MMEJ competes with BIR in the repair of telomeric DSBs and implies that the loss of some potential genes important for BIR, such as POLD3 or POLD4, may promote MMEJ

In summary, during the selection of a DSB repair pathway, MMEJ is not only a backup pathway but also an essential pathway contributing to genome instability, especially in telomeres. However, as MMEJ and HR are both active in the S phase because of the initial resection stimulation by CDK phosphorylated CtIP function, it could be that the most dramatic synthetic lethal effect is between MMEJ and HR or BIR. For competition between NHEJ and MMEJ, a less significant synthetic lethal effect might be observed; however, such an effect might be possible if cells lack NHEJ but mainly rely on MMEJ to repair DSBs in the G1 phase. It could be very important to explore the synthetic lethal relationship between MMEJ and other pathways, which could explore the multifunctional mechanism and new role for proteins located on DSBs.

## 5. Connection between MMEJ and Replication Stress

As CDK-mediated end resection initiation mainly occurs in the S phase, MMEJ might be more active in the S phase and compete with HR. It has also been reported that CDK1/Aurora A and PLK1 can phosphorylate CtIP and that PLK1 can phosphorylate CtIP at serine 327 to promote MMEJ and inactivate the G2/M checkpoint [60]. Competition between HR and MMEJ could start from the initial resection, but dramatic differences may be observed in extensive end resection. HR has been better studied regarding the mechanism coupled with DNA replication stress. Here, I mainly aim to discuss the potential connection between MMEJ and replication stress, and there is still much unknown regarding this question. Replication-coupled repair is defined as a mechanism that processes damaged DNA in coordination with the replisome and maintains genome stability. Double-strand break repair protects stalled forks from degradation and restarts broken forks. HR restores replication upon DNA breaks that occur at stalled forks [89]. Replication machinery can bypass some forms of damage and postpone lesion removal until after DNA synthesis is complete, and these are essential functions needed in every cell division cycle [90]. Accurate replication of DNA requires stringent regulation to ensure genome integrity. Whether and how MMEJ helps replication fork protection or restarts broken forks is a new question. Some evidence linked to MMEJ in replication stress is currently limiting the study of PARP function. Inhibition of poly (ADP-ribose) polymerase (PARP) increases the speed of fork elongation and does not cause fork stalling [91], which is in contrast to a previous model in which PARP inhibitors induced fork stalling and collapse [92]. This finding raises the question about PARP function in stimulating initial resection, but blockade of BRCA-2-mediated extensive resection might be a mechanism to promote MMEJ rather than HR. Therefore, in the absence of PARP1, spontaneous single-strand breaks (SSBs) cause the collapse of replication forks and trigger HR. Some evidence from previous studies showed that PARP1 is required for SSB repair in the G1 phase but not the S phase in the cell cycle, and PARP1-dependent and independent SSB repair pathways exist [93]. However, it has been also reported that PARP1 knocked down elicited hyper-radiosensitivity but PARP inhibitor-induced radiation sensitivity occurs only in those cells treated in the S Phase [94].Other studies suggested that the synthetic lethality induced by PARP inhibitors is not due to the inhibition of SSB repair [95]. This evidence implies that PARP inhibitor sensitivity may be mediated by other mechanisms in addition to inhibiting SSBR [93] PARP inhibitor-induced collapsed replication forks cannot be repaired by NHEJ, leading to death in HR-deficient cells [96]. It has been also reported that PARP inhibitors cause defects in ssDNA gap filling during DNA replication and loss of nuclear DNA ligase III resensitizes HR-restored BRCA1-deficient cells to PARPi by exposing post-replicative ssDNA gaps [97].

On the other hand, I speculate that it is presumably possible that PARP-promoted MMEJ could repair the collapsed replication fork and restart broken forks, which might also affect the limited resection of MMEJ to inhibit MMEJ and process single-strand gap fill-in by PARP inhibitors. This mechanism is not yet fully understood (Figure 5).

## 6. Microhomology-Mediated Break-Induced Replication (MMBIR)

It is also interesting to discuss the microhomology function to connect with replication fork stalling. Similar to BIR as described before, this mechanism was proposed to explain telomere maintenance in yeast and human cell lines that have lost telomerase activity [99]. It is usually considered an accurate process because repeated invasions are strongly RecA/Rad51-mediated and involve long lengths of homology (approximately 50 bp in *E.*
*coli* [100] and more in eukaryotes [101,102]) between DNA sequences. BIR at common fragile sites occurs after MUS18 cleaves unreplicated DNA during mitosis, and the restarting of stalled replication forks could be RAD51-dependent or independent [90]. Interestingly, a previous study suggested a novel pathway of the microhomology-mediated BIR (MMBIR) [103]. The study reported that complex genomic rearrangements (CGRs) appeared to result from MMBIR, which is a replicative mechanism involving template switching at the microhomology position. In yeast, a collapse of homology-driven break-induced replication is caused by defective repair DNA synthesis in the absence of Pif1 helicase, which leads to template switches involving 0–6 nt of homology, followed by resolution of recombination intermediates into chromosomal rearrangements. In humans, PIF1 helicase promotes BIR, especially at broken replication forks. The mechanism of PIF1-dependent BIR is used for homology-initiated recombination requiring long track DNA synthesis. In addition, PCNA-dependent loading of PIF1 on the broken forks is critical for BIR activation. Loss of PIF1 is synthetically lethal with loss of FANCM, which is involved in protecting common fragile sites-induced BIR [104]. So, the absence of PIF1 is likely to help switch BIR to MMBIR. In addition, the study also showed that MMBIR is driven by the translesion synthesis (TLS) polymerases Polζ and Rev1. Translesion polymerase bypasses DNA damage lesions during DNA replication. If a lesion is not repaired or bypassed, the replication fork can stall, possibly leading to cell death. An interruption of BIR with fully homologous chromosomes in yeast triggers a switch to MMBIR catalyzed by TLS polymerase [105] (Figure 6). In human cells, TLS polymerases enable the bypass of replication fork-stalling lesions in a potentially error-prone manner, given their low fidelity. TLS-mediated lesion bypass is thought to occur in two steps involving the insertion of a DNA base opposite the lesion by Y-family DNA polymerases (Polη, Polι, Polκ, and REV1), followed by the extension of DNA synthesis by the B-family Polζ complex (REV3L/REV7/POLD2/POLD3) [106,107]. Notably, replication-associated DNA damage induces the monoubiquitination of proliferating cell nuclear antigen (PCNA) on lysine 164 (K164) by RAD18/RAD6, resulting in the binding of TLS DNA polymerases to PCNA [108]. Coordination of TLS activities is mediated by REV1, which is a TLS scaffold protein that binds PCNA through its N-terminal BRCA1 C-terminus (BRCT) domain and Polη, Polι, and Polκ or the REV7 subunit of the Polζ complex through distinct C-terminal interaction surfaces [109,110,111,112]. These mechanisms potentially indicate that the REV7 subunit of the Polζ complex and Y-family DNA polymerases (Polη, Polι, Polκ, and REV1) could also be functional parts of MMBIR. Interestingly, in addition to restarting DNA synthesis at stalled forks, TLS polymerases also fill in single-strand DNA (ssDNA) gaps remaining after DNA replication. Post-replicative repair of ssDNA gaps can also occur through HR-mediated processes, a pathway dependent on PCNA polyubiquitination that uses the sister chromatid as a template for DNA synthesis. More evidence indicates that TLS and HR can act as alternative compensatory processes for ssDNA gap repair, which also provides the logical hypothesis that PCNA may be a key functional factor in MMBIR and help the TLS polymerase family bypass replication fork-stalling lesions, especially for ssDNA gaps, as a proposed mechanistic model. It could also be interesting to investigate PCNA regulation, including the role of p21, which directly binds to PCNA through the C-terminal region (aa 139–164, containing the PIP box QTSMTDFY (aa 144 to 151)) and promotes CDK4/6 but blocks CDK2 [113]. When p21 degradation is prevented in the context of high PCNA ubiquitination, the recruitment of specialized polymerases to replication factories is impaired. The stable p21-PCNA interaction is important for p21 in TLS inhibition [114]. In addition, Pol k is retained on DNA by a secondary interaction between the C-terminal domain containing two ubiquitin-binding zinc fingers (UBZs) and the ubiquitins flexibly conjugated to PCNA when the internal PIP-box interaction is lost [115]. Trimeric PCNA is subject to monoubiquitylation via the activity of multiple E3 ligases (CRL4-CDT2) and subsequent replication stress via Rad18 [116]. Therefore, according to the potential mechanism of the interaction between PCNA and TLS polymerases in promoting MMBIR during ssDNA gaps, I predict that microhomology-mediated end joining likely helps the re-establishment of replication fork stalling and restart replication to support better DNA damage lesion repair.

A previous study also showed re-replication, which is a repeated activation of the same DNA replication origin that also generates DSBs at sites of fork collisions and can lead to genome instability. Resection-mediated repair, such as HR and MMEJ repair, is promoted by CDK activity and used during the S/G2 phase, consistent with the timing of re-replication events. Interestingly, the McVey group reported that loss of DNA polymerase θ (Pol θ) impedes the progress of re-replication forks at a specific genomic locus [117]. Polymerase θ/mutagen-sensitive 308 (*mus308*) displays normal origin firing but reduced fork progression at two regions of re-replication, and MMEJ compensates for the loss of NHEJ to repair re-replication DSBs in a site-specific manner. Fork progression is enhanced in the absence of *Drosophila* Rad51 homologs, which could lead to loss of HR activity, but switch to MMEJ to repair after re-replication in human cells. So, it cannot repair all re-replication-induced DSBs [117,118].

## 7. Therapeutic Opportunities of New Regulators of MMEJ for Cancer Treatment

MMEJ was thought to be a backup pathway to repair DSBs. However, its potential clinical usage based on the synthetic lethality principle between MMEJ and HR has attracted more attention [37]. Specific targeting of MMEJ for important genes in different cancer types to regulate DSB repair pathways is a promising and novel approach for cancer therapy, such as the utilization of PARP inhibitors. In addition, new POLQ inhibitors have also been studied as promising drugs in cancer treatment, including known cancer drivers such as BRCA1/2 [119]. POLQ inhibitors could also elicit and target PARP inhibitor-resistant BRCA1/2 mutant cancer [120].

Recently, along with my co-authors, I reported that retinoblastoma protein (RB), as a classical tumor suppressor, is functional in the selection of DSB repair pathway; in particular, RB-deficient human cancer cell lines use MMEJ as the major pathway to repair DSBs [61]. Interestingly, RB-deficient cancer is sensitive to the PARP inhibitor olaparib by targeting MMEJ as a novel mechanism for PARP inhibitor sensitivity. Combining a PARP inhibitor and etoposide enhances the killing effect in RB-defective cancers [121]. It could potentially be used for retinoblastoma treatment. RB inactivation has been shown in many cancer types, not only retinoblastoma but also melanoma (including uveal melanoma, usually with higher expression of phosphorylated RB) [122,123], small cell lung cancer [124], and approximately 30% of triple-negative breast cancers [125]. Recently, another study also showed that *RB1* mutant osteosarcoma is sensitive to PARP1,2 inhibition, which is not associated with canonical homologous recombination defect signatures but is accompanied by the activation of DNA replication as a prerequisite for sensitivity. The sensitivity of PARP inhibitors with a background of RB1 loss surpasses that seen in BRCA-mutated backgrounds and could establish a clinical benefit of PARP inhibitors [126].

I would like to speculate that targeting MMEJ, such as with POLQ inhibitors, would be an exciting and novel treatment for all RB-loss cancers. Furthermore, homologous recombination-defective cancer might be highly likely to rely on NHEJ or MMEJ for repairing DSBs, which can also be investigated, as loss of the genes promoting HR could be sensitive to POLQ inhibitors with possibilities in clinical use. On the other hand, CDK is an essential factor for DNA end resection, which is the first step to promoting MMEJ. It might be possible that the mechanism of CDK inhibitors as cancer therapy could also partially target MMEJ; however, further studies may need to follow up with those proposed models. The most beneficial part of targeting MMEJ in the selection of DSB repair pathways is that it causes less damage to normal cells and has a specificity for cancer cells that is comparable to chemotherapy drugs. The killing effect may also be enhanced by combination with chemotherapies such as etoposide or cisplatin. It could dramatically decrease chemotherapy use and secondary cancer prevalence induced by DNA damage. This could be the future direction for developing specific MMEJ-targeting inhibitors as a novel angle for cancer therapy.

In addition to *RB1*, another interesting gene, *MYCN*, which is a member of the *MYC* family of oncogenes, is important for its transcriptional acidity [127]. Deregulation of MYCN occurs in both pediatric cancers and adult cancers. Amplification of the MYCN oncogene is present in many types of pediatric cancer, including 18–20% of neuroblastomas [128,129], 25% of alveolar rhabdomyosarcomas [130], 5–10% of medulloblastomas [131], and <5% of retinoblastomas [132]. In adult cancers, amplification of MYCN *is* present in 40% of neuroendocrine prostate cancers and 5% of prostate adenocarcinomas [133], 15–20% of small-cell lung cancers [134], and 17.5% of basal cell carcinomas [135]. Overexpression of MYCN *is* also present in breast cancer, lymphoblastic leukemias, and glioblastoma [127]. Overexpression of MYCN interferes with FBXW7-mediated degradation, leading to MYCN stabilization [136]. Dephosphorylation of MYC-S62 via protein phosphatase 2A (PP2A) enables E3 ligase FBXW7 binding to phosphorylated MYC-T58, targeting it for ubiquitination and subsequent degradation by the proteasome [137]. Interestingly, a previous study found that MYCN amplifies neuroblastoma survival by the MMEJ mechanism to overcome DSBs [138]. Inhibition of ligase III and PARP1 leads to neuroblastoma cell death [138]. A similar effect was also found in another group, in which knocking down the MMEJ component by siRNA reversed MYCN effects on neural crest stem cell (NCSC) proliferation and uncovered the link between MYCN and MMEJ expression in neuroblastoma DNA maintenance and developmental tumor initiation [139].

Targeting MMEJ in specific gene mutation cancers is a very promising approach for cancer development and novel therapy. Further understanding of the mechanism by which these genes regulate microhomology use will also be very interesting to explore and may contribute to understanding genome instability.

## Figures and Tables

**Figure 1 ijms-23-12937-f001:**
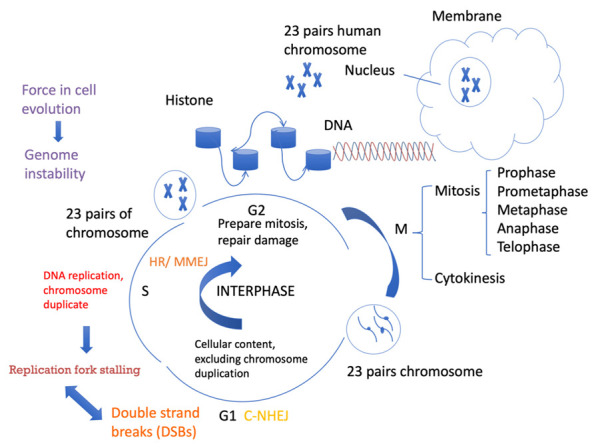
Cell cycle progression and DSB repair. Cells spend a long time in interphase and a very short amount of time in mitosis. In the G1 phase, cellular contents, excluding chromosomes, are duplicated. In the S phase, DNA is replicated. In the G2 phase, cells need to be prepared for mitosis. Different DSB repair pathways favor different phases of the cell cycle. C-NHEJ can take place anytime during the cell cycle, but it is more active in G1. HR requires sister chromatids as templates for repair and mainly occurs in the S/G2 phases. MMEJ activity also increases in the S phase and can also occur in the G1 phase.

**Figure 2 ijms-23-12937-f002:**
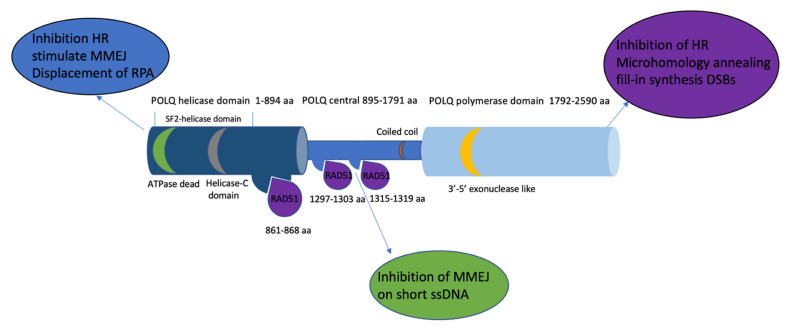
The POLQ protein includes a C-terminal domain containing a polymerase domain (1792–2590 aa, light blue), an N-terminal domain containing a helicase domain (1–894 aa, navy blue), and a central domain (895–1791 aa sky blue). The RAD51 (purple) binding site is located in both the helicase domain and central domain (861–868 aa, 1297–1303 aa, and 1315–1319 aa). The 3′-5′ exonuclease-like region (yellow) is located on the polymerase domain. The superfamily 2 (SF2) helicase domain contains a conserved death box motif (green) and a conserved helicase C-terminal domain (grey). Text bubbles (blue, purple, and green) indicate the function of each domain of POLQ.

**Figure 3 ijms-23-12937-f003:**
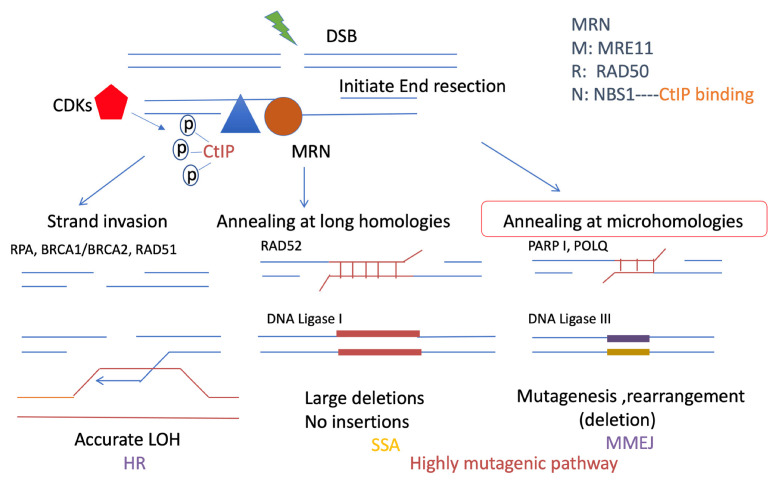
Resection-mediated DSB repair pathways. Resection is important for promoting DSB repair pathways, including HR, SSA, and MMEJ. They all share a common initial resection procedure, which is promoted by the MRN complex: RAD50, NBS1, MRE11 (brown), and CtIP (blue). In particular, CtIP is phosphorylated by CDKs (red) and stimulates the MRN complex to initiate DNA end resection, which is a short range of ssDNA overhangs. HR and SSA both require extensive resection, which requires a long range of resections. This resection event is different from the resection that promotes MMEJ. MMEJ and SSA are both highly mutagenic pathways. However, MMEJ requires microhomologies usually less than 25 bp, and is promoted by PARP1 and POLQ, which ligate the DSB ends by ligase III, with mutagenesis, deletions, and rearrangement. By contrast, longer microhomology will require RAD52, which is necessary for SSA, which could generate large deletions. HR is more accurate than MMEJ and SSA. It causes strand invasion instead of using the sister chromatid after extensive resection. RPA is required for extensive resection and can be replaced by RAD51 with the support of BRCA2.

**Figure 4 ijms-23-12937-f004:**
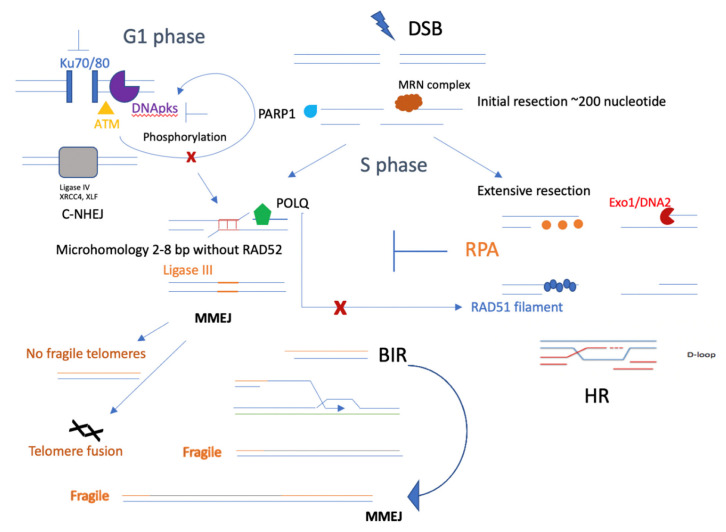
Competition between MMEJ and other DSB repair pathways. MMEJ, as an alternative nonhomologous end-joining pathway, competes with C-NHEJ, especially when the KU family is defective. It could also be possible that losing DNApk subunits or blocking ATM phosphorylation of DNApk would lead to defective C-NHEJ but elevated MMEJ. In the S phase of the cell cycle, initial resection will be the key step in promoting both HR and MMEJ, and the synthetic lethal relationship between HR and MMEJ could mainly occur in the S phase. In addition, several important factors, such as PARP1, may contribute to initial resection and promote MMEJ, and POLQ could functionally inhibit RAD51 [37] filaments, leading to less HR. In telomeres, MMEJ helps to repair most DSBs and does not generate fragile telomeres; however, it does cause telomere fusion. Instead, BIR would form fragile telomeres, which could be repaired better by MMEJ factors, such as PARP1 and ligase III.

**Figure 5 ijms-23-12937-f005:**
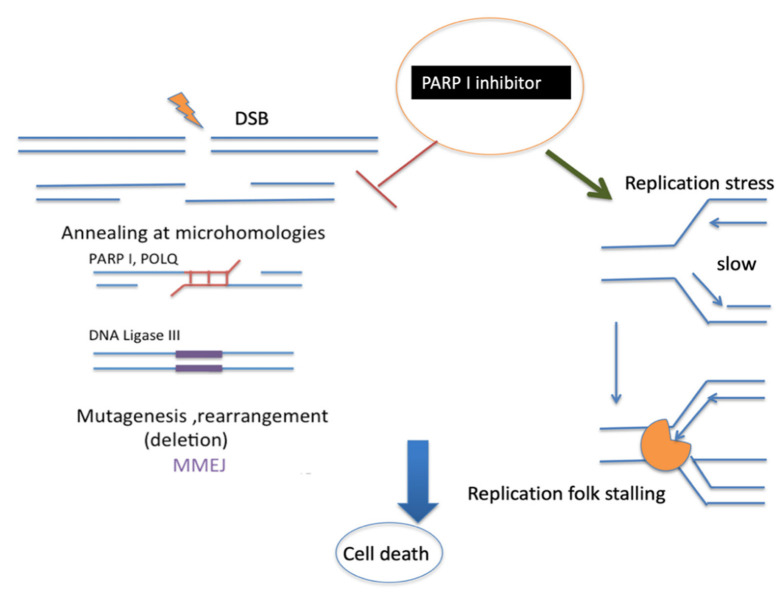
Proposed mechanism of PARP1 inhibitor. PARP1 is well known to promote MMEJ at the early stage. This is followed by POLQ and DNA ligase III activity. PARP1 inhibitors can also stabilize replication fork stalling [98] with polymerase (orange). Both mechanisms of PARP1 inhibition could induce cancer cell death.

**Figure 6 ijms-23-12937-f006:**
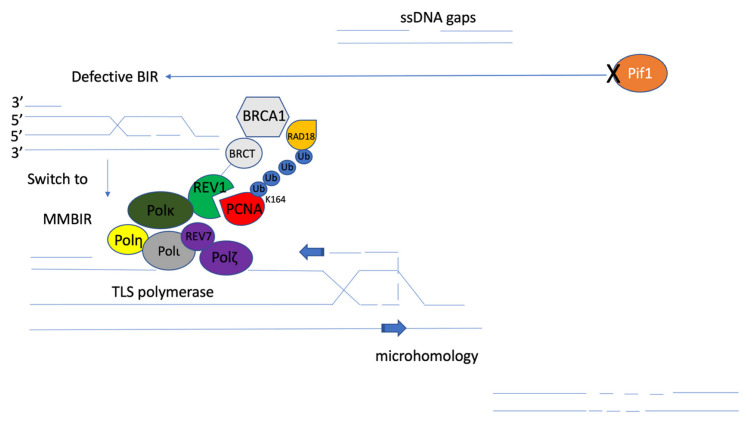
Proposed mechanistic model of microhomology-mediated break-induced replication (MMBIR). The disfunction of Pif1 helicase (orange) leads to template switches involving microhomology from defective BIR. MMBIR is driven by the translesion synthesis (TLS) polymerases Polζ and Rev1 [105]. Monoubiquitination of PCNA (red) on lysine 164 (K164) by RAD18/RAD6 (yellow) results in the binding of TLS polymerases to PCNA. TLS activities are mediated by REV1 (green) binding to PCNA through its BRCT domain (grey), and Polη (bright yellow), Polι (dark grey), Polκ (dark green), or the REV7 subunit of the Polζ (purple) complex through distinct C-terminal interaction surfaces through mutagenic repair of primpol-dependent ssDNA gaps [8]. That mutagenic repair could be microhomology-mediated repair, which will support the reestablishment of replication fork stalling and DNA damage lesion bypass.

## Data Availability

Not applicable.

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
