# Peer review of "Contribution of Microhomology to Genome Instability: Connection between DNA Repair and Replication Stress"

_ijms, 2022, doi:10.3390/ijms232112937_

Round 1
Reviewer 1 Report
In the manuscript “Contribution of microhomology to genome instability: Connection between DNA repair and replication stress” by Jiang, the author reviewed recent developments in understanding the role of Microhomology-mediated end joining (MMEJ) in repairing double-strand breaks and genome stability maintenance.
There are only two comments:
1. On page 2, line 60, “It is initiated by DSBs but also addresses replication fork stalling-induced 60 DSBs.”
HR is also used to repair single-stranded gaps, besides DSBs.
2. On page 3, line 96, “There are four types 96 of HR include gene conversion, synthesis-dependent strand annealing (SDSA), break-in-97 duced replication (BIR), and single strand annealing (SSA).”
SSA is not classified as a sub-type of homologous recombination (HR).
Author Response
Thank you very much for the kind comments and suggestions.
- On page 2, line 60, “It is initiated by DSBs but also addresses replication fork stalling-induced DSBs.”
HR is also used to repair single-stranded gaps, besides DSBs.
Reply: Thank you very much for this comment. I have changed the sentence as:” It is initiated by DSBs but also addresses replication fork stalling-induced DSBs and acts as an alternative process for ssDNA gaps repair[8].”
- On page 3, line 96, “There are four types of HR include gene conversion, synthesis-dependent strand annealing (SDSA), break-induced replication (BIR), and single strand annealing (SSA).”
SSA is not classified as a sub-type of homologous recombination (HR).
Reply: Thank you very much for this correction. I have revised it as: “There are some types of HR including gene conversion (GC), synthesis-dependent strand annealing (SDSA), and break-induced replication (BIR). Another DNA DSB repair pathway is single-strand annealing. They all initiate repair with the resection of DSB ends by a 5’-to-3’ exonuclease to produce long 3’-ended, single-stranded DNA (ssDNA) tails[14]”.
Reviewer 2 Report
Review written by Jiang describes the involvement of microhomology to genome instability. Manuscript is nicely written and easy to follow.
I found the Chapter entitled Terapeutic opportunities of new regulators of MMJ for cancer treatment very interesing, however I would suggest to discuss this topic even more detail (PMID: 15829967, PMID: 24202391, ).
Additionally, I have some minor comments:
1. Line 58 Explain the abbreviation HR
2. Sometimes Author writes ligase III and sometimes uses capital letter, be consistant, please
3. Saccharomyces cerevisiae should be written with italic
4. Line 453 and 455: ...amplification/overexpression of MYCN is... is should not be written with italic
Author Response
Thank you very much for all of the kind corrections and suggestions.
- Line 58 Explain the abbreviation HR
Reply: I have explained the abbreviation of HR by “Homologous Recombination (HR).”
- Sometimes Author writes ligase III and sometimes uses capital letter, be consistant, please
Reply: I have changed all “ligase” to “Ligase”.
- Saccharomyces cerevisiae should be written with italic
Reply: I have changed it to italic.
- Line 453 and 455: ...amplification/overexpression of MYCN is... isshould not be written with italic
Reply: I have changed “MYCN” to non- italic.